# Advances and Current Status in the Use of Cuticular Hydrocarbons for Forensic Entomology Applications

**DOI:** 10.3390/insects16020144

**Published:** 2025-02-01

**Authors:** David Stewart-Yates, Garth L. Maker, Stefano D’Errico, Paola A. Magni

**Affiliations:** 1School of Medical, Molecular and Forensic Sciences, Murdoch University, Murdoch, WA 6150, Australia; bobby.stewart-yates@murdoch.edu.au (D.S.-Y.); g.maker@murdoch.edu.au (G.L.M.); 2Department of Medical Surgical and Health Sciences, University of Trieste, 24149 Trieste, Italy

**Keywords:** blow flies, chemical analyses, species identification, age estimation, PMI, environmental factors

## Abstract

Cuticular hydrocarbons present a valuable tool in forensic entomology, aiding species identification and age estimation of necrophagous insects, particularly blow flies. This review provides a detailed overview of recent advancements in the applications of cuticular hydrocarbons, such as in post-mortem interval estimation. Additionally, it explores the factors contributing to intra-species variation, including age, sex, temperature, and geographical origin, and details how these variations can provide additional insight during legal investigations. While promising, challenges remain in the use of cuticular hydrocarbons in forensic investigations, and further research is required to enhance reliability of this method.

## 1. Introduction

Medico-legal (forensic) entomology is the discipline focused on the use of insects and other arthropods associated with animals and humans, whether alive or at various stages of decomposition, as part of a legal investigation [1]. This field plays a crucial role in estimating time of death (post-mortem interval, PMI or minPMI [2]), investigating cases of abuse or neglect, providing insight into ante-mortem body injury, and identifying post-mortem movement of a body, as well as detecting the presence of toxic substances or foreign DNA [3]. By analysing insect evidence, forensic entomologists can offer valuable insights that contribute to solving crimes and legal disputes. Among the necrophagous entomofauna in a terrestrial temperate environment, blow flies (Diptera: Calliphoridae) are typically the first insects to colonise a body, and their predictable life cycle and widely researched ecology and behaviour are essential for their forensic application [4,5]. In a criminal investigation, forensic entomologists collect or receive specimens collected by proxies [3], identify the species, and use relevant micro- and macro-environmental data to age them. They also consider case-specific ecological information and the overall entomological assemblage to estimate a colonisation interval, which can be correlated to the cadaver PMI [6].

Blow fly species identification and ageing can be achieved through various methods; the most common are morphological and biomolecular, with physical and chemical methods increasingly recognised as complementary techniques. In morphological analysis, characteristics of blow fly eggs, larvae, pupae, and empty puparia are examined under a microscope and compared to published dichotomous keys. This approach is also used to determine the specimen’s life stage by observing specific features that change over time, such as posterior spiracles and mouthparts [7]. An important limit of the morphological method is the requirement of personnel with taxonomical expertise. Alternatively, DNA-based methods can be used for species identification as a primary technique or to confirm morphological identification [8,9]. DNA-based methods can also serve as an alternative to morphological analyses in cases where specimens have been altered by physical damage or toxins [9,10,11]. Additionally, molecular methods have been explored for ageing specimens by observing changes in gene expression during development, which may serve as age estimation markers; however, studies in this area remain limited [12]. Within physical methods, reflectance spectroscopy [13,14] and infrared spectroscopy [15] can be applied for both species identification and ageing, while computed tomography (CT) scanning has been successfully employed to age intra-puparial forms [16,17,18]. Although promising, these methods are constrained by the limited number of studies and the frequent requirement for sophisticated technology that is not universally accessible.

Chemical methods for the identification and ageing of forensically important insects rely on the analysis of cuticular hydrocarbons (CHCs), which are compounds that form the wax layer on the insect’s cuticle surface (epicuticle) (Figure 1) [19,20].

These CHCs are species-specific, present at all stages of an insect’s life, and vary throughout its life cycle. The extraction process of CHCs does not require specialised personnel and the analysis is primarily carried out by gas chromatography–mass spectrometry (GC-MS) [21], using an instrument available in most forensic or chemistry laboratories. The CHC technique shows promise, with a growing body of research published in the field [19,20,22].

To date, three reviews summarising the use of CHCs in forensic entomology have been published, in 2015 [20], 2017 [19], and 2021 [22]. While these reviews were comprehensive, the current review differs significantly by including a dedicated section on the factors influencing CHC profiles and how these factors can be leveraged to improve the method’s reliability and applicability—an aspect not addressed in earlier reviews. Notably, this review also incorporates research published since 2021, a period during which substantial advancements have been made in understanding intra-specific CHC variation.

The present review provides an updated perspective that not only summarises recent advancements but also critically evaluates factors influencing the CHC profile, offering investigators practical insights for employing this technique in forensic applications.

## 2. Discussion

### 2.1. CHC Overview, Extraction Method, and Analytical Procedure

The integument of an insect consists of a single epidermal cell layer and the cuticle, which is comprised of three layers: the endocuticle, exocuticle, and epicuticle (Figure 1). The epicuticle is the outermost layer of the insect cuticle, and in blow flies it represents the external surface of the body at every stage of life: egg, larva, puparium, and adult. This layer serves as a protective barrier, preventing desiccation and providing resistance to environmental stressors [23]. It contains cuticular hydrocarbons (CHCs), which play a vital role in communication, protection, and hydrophobicity [21]. A visual representation of the insect integument, showing the location of CHCs typically used in chemical analyses and list of common CHCs in blow flies, is provided in Figure 1.

The CHCs of an insect can include saturated or unsaturated hydrocarbons, and may have one or more methyl groups attached [24]. The saturated form (*n*-alkanes) consists of carbon chains joined by single bonds, while unsaturated forms (olefins) contain one (alkenes or monoenes), two (alkadiene or dienes), or three (alkatrienes or trienes) double bonds between carbon atoms along the chain [24]. Additionally, olefins can exist in two possible isomeric forms referred to as the cis- (*Z*-alkenes) or trans-forms (*E*-alkenes); however, all insect CHCs identified so far are in the *Z*-configuration [24]. As a result, a typical insect CHC profile includes *n*-alkanes, *Z*-alkenes, and methyl-branched alkanes [25] within the carbon chain length from 19 to 35 (C_19_–C_35_) [26].

The methodology outlined by Moore [25] is the preferred approach for CHC studies, with extraction of CHCs commonly performed using liquid–liquid extraction techniques and a non-polar solvent such as hexane or pentane. Insects are fully submerged in a solvent (typically 350–500 μL) within a GC vial for 10 min. The number of insects per vial depends on the species and life stage. For example, 20–30 specimens are suggested for blow fly eggs and first instar larvae, 5–15 for second instar larvae, 2–3 for third instar larvae, and 2 each for post-feeding larvae, full puparia, and empty puparia. A single adult is sufficient to provide adequate CHC concentration for GC-MS analysis. Following the 10-min extraction, the solvent containing the CHCs is collected into a clean vial and transferred to a micro-insert to dry down completely. This drying step removes impurities and concentrates the CHCs. The extracted CHC profile is then analysed and quantified using GC-MS, providing detailed information on the composition and relative abundance of the hydrocarbons. Finally, multivariate statistical techniques, such as principal component analysis (PCA) are applied to data obtained from GC chromatograms to aid in trend visualisation and interpretation.

### 2.2. CHCs as a Tool for Species Identification

Chemotaxonomy is a branch of taxonomy that utilises the chemical composition of an organism to classify and differentiate between species [27]. In entomology, chemotaxonomy relies on the analysis of various chemical markers, such as CHCs, along with other chemical components such as proteins, amino acids, fatty acids, volatile organic compounds (VOCs) and also molecular markers including DNA and RNA [28,29,30]. These chemical markers are often species-specific and reflect an insect’s genetic makeup and physiological traits, contributing to the differentiation of closely related species or populations through variations in composition, concentration, and the presence or absence of specific compounds [31]. As heritable components of the insect metabolome, CHCs provide insights into an insect’s genotype and also its environmental interaction, making chemotaxonomy a powerful tool for classification and the study of insect biodiversity and evolution [21]. The CHC profile can consist of only a few to hundreds of compounds offering numerous combinations that facilitate species delimitation [31].

The first use of the CHC technique for identifying necrophagous flies was published in 2007 by Ye et al. [32]. This study revealed distinct CHC profiles among the empty pupal cases of several necrophagous fly species, including *Aldrichina grahami* (Aldrich) (Diptera: Calliphoridae), *Chrysomya megacephala* (Fabricius), *Lucilia sericata* (Meigen), *Achoetandrus rufifacies* (Macquart), *Boettcherisca peregrina* Rohdendorf (Diptera: Sarcophagidae), and *Parasarcophaga crassipalpis* (Macquart). Building on this work, Roux et al. [33] became the first to examine the complete ontogeny of three forensically relevant blow fly species—*Calliphora vomitoria* (L.), *C. vicina* Robineau-Desvoidy, and *Protophormia terraenovae* Robineau-Desvoidy—using GC-FID and discriminant analysis. Their results demonstrated the potential to distinguish both species and developmental stage (egg, larvae, post-feeding larvae, full puparia, and adults) based on CHC profile.

Following these advances, Moore et al. [34] demonstrated the utility of CHC profiling for identifying three forensically significant blow fly species (*L. sericata*, *C. vicina*, and *C. vomitoria*) at the first instar larvae, bypassing the need for rearing to later developmental stages for reliable morphological identification. In their findings, the CHC profiles of the two *Calliphora* species were more similar in their profile compared to *L. sericata*, as anticipated; however, significant differences in the abundance of methyl-branched alkanes facilitated differentiation. For instance, *L. sericata* exhibited only three methyl-branched alkanes, whereas *C. vicina* and *C. vomitoria* displayed 20 and 10, respectively. Additionally, some compounds proved to be species-specific, such as 12 + 14-methylhexacosane in *C. vicina*, 2-methylhexacosane in *C. vomitoria* and octadecane in *L. sericata*. Other compounds, like 9 + 11-methylpentacosane were present in all three species but varied in concentration, with the highest concentration found in *C. vomitoria* and lowest in *L. sericata*.

Barbosa et al. [35] analysed wild caught adult *L. cuprina* (Wiedemann)*, Cochliomyia macellaria* (F.), and *Hemilucilia segmentaria* (F.) in Brazil, revealing CHC profiles that allowed for both species and sex differentiation. In another study, Lunas et al. [30] identified chemical differences in CHCs of eggs of *Ch. megacephala, Ch. albiceps* (Wiedemann)*, L. eximia* (Wiedemann), and *L. cuprina*, enabling species distinction at the egg stage. Zaher et al. [36] conducted the first study on cuticular hydrocarbons for blow fly species identification in Egypt, successfully distinguishing between *L. sericata, Ch. Albiceps*, and *Ch. marginalis* (Wiedemann), while Paula et al. [37] established the chemotaxonomic profiles across all developmental stages (eggs, first, second, and third instar larvae, pupae, and adults) of *Ch. megacephala*.

Beyond the Calliphordae family, significant studies by Braga et al. [38], Moore et al. [39], and Zhang et al. [40] focused on the challenging Sarcophagidae family. These studies demonstrated distinct CHC profiles for various life stages of species such as *Peckia (Peckia) chrysostoma* (Wiedemann), *Peckia (Pattonella) intermutans* Walker, *Sarcophaga (Liopygia) ruficornis* (F.), and *Sarcodexia lambens* (Wiedemann) [38]. The results showed that the insects clustered separately on a hierarchical tree based on their CHC profiles using Bray–Curtis similarity index. Finally, in a study on the stability of CHCs over time, Moore et al. [39] confirmed the feasibility of identifying eleven Sarcophagidae species more than 100 years old, underscoring the long-term reliability of CHCs as taxonomic markers.

CHCs offer a promising method for species identification in entomological evidence across all stages of development and are particularly valuable for older or damaged specimens. In cases where traditional methods lack sufficient precision, CHCs can serve as an alternative or confirmatory tool, even for cryptic species where morphological and DNA-based approaches have previously failed.

### 2.3. CHCs as an Ageing Tool

Accurate ageing of insects is crucial in forensic entomology for estimating colonisation intervals and the associated PMI. Morphological methods, while effective with specialised personnel, can become unreliable when samples are degraded or incomplete. Advanced technologies, such as CT scanning and molecular methods, show potential but are often restricted to certain life stages and are hindered by high costs and limited research. In this context, CHCs have emerged as a promising tool for age estimation across various insect life stages.

Roux et al. [33] were the first to complete a comprehensive hydrocarbon analysis across the blow fly life cycle, identifying distinct profiles for each life stage and noting a trend from low to high molecular weight CHCs. Subsequent studies by Zhu et al. [41], Moore et al. [42,43], Xu et al. [44], and Sharma et al. [45] have further documented an increase in high molecular weight hydrocarbons as blow fly larvae age. This shift is likely due to the extra waterproofing requirement as the larvae moves from their food source to a drier environment for pupation.

Using CHCs and artificial neural networks, Moore et al. [43] achieved age estimations for *C. vicina* and *C. vomitoria* larvae with more than 87% accuracy. Shang et al. [46,47] investigated the use of cuticular hydrocarbons in pupal ageing of *Sarcophaga peregrina* Robineau-Desvoidy, presenting promising results. Additionally, Zhang and colleagues [40] explored CHCs across the full life cycle of *S. peregrina* and were able to distinguish larval days 1–6, pupae days 13 and 15, and adult days 16–25 (sampled every 48 h) using orthogonal partial least squares discriminant analysis (OPLS-DA). However, pupal form days 7, 9, and 11 tended to cluster together, highlighting some limitations in resolution at certain stages and underscoring the need for further research to enhance the accuracy of this technique.

Several authors have explored the weathering or degradation of CHCs in empty pupal cases and their potential application in forensic science [48,49,50,51,52,53,54,55,56]. Weathering refers to the process by which the chemical components of the chitinous exoskeleton, such as CHCs, undergo gradual change due to environmental factors like temperature, humidity, and exposure to UV light. This degradation results in alterations to the type and concentration of CHCs extracted, which can serve as a timeline of decomposition. Pupal cases can remain in the environment for extended periods; however, their morphological examination alone cannot provide an estimate of the duration. In these cases, ecological studies can offer indications of this timeframe. Studies have presented time-dependent changes in the CHC profiles which can be used to estimate the weathering duration, and subsequently, the PMI. Zhu et al. [49] observed significant chromatographic changes in hydrocarbon profiles of *Ch. megacephala* puparia over 90 days of weathering, displaying a predictable change that was found to depend on the chemical structure and molecular weight of hydrocarbons. Moore et al. [56] also successfully characterised the hydrocarbon profiles of the empty puparia of *L. sericata* and *C. vicina* over a nine-month period.

Ageing adult flies poses unique challenges, as morphological changes occur primarily within the first few hours after eclosion from the puparium [57]. Pechal et al. [58] demonstrated that the hydrocarbon profile of adult female *C. macellaria* and *Ch. rufifacies* varied with age over a 30-day study period. Braga et al. [59] reported the *n*-alkane *n*-C_29_ was the most abundant in both male and female *Ch. putoria* across all ages; however, considerable age- and sex-related differences in the profiles allowed for age estimation across a five-day period. Moore et al. [57] sampled adult *C. vicina* and *C. vomitoria* at 1, 5, 10, 20, and 30 days post-emergence, and found days 1, 5, and 10 could be distinguished by PCA. *L. sericata* were sampled 1, 5, and 10 days post-emergence and were all distinguishable. An isolated study of only *n*-C_25_ found its abundance to increase significantly and linearly with increased adult age over 20 days [60].

An understanding of how CHC components change with age and identifying trends may provide a complementary tool for age estimation of insect evidence, particularly for life stages that are challenging to examine by morphology, such as pupae and empty puparia. Inside the puparia, pupae undergo significant transformations that can only be appreciated after the puparia is carefully removed by a trained entomologist or through the application of specific technologies [13]. Additionally, empty puparia often lack obvious external features, and their condition can be highly affected by weathering [61], making accurate age estimation even more difficult using traditional methods.

### 2.4. Factors Influencing CHC Composition

The composition of CHCs in insects is not static; rather, it is influenced by a complex interaction of both genetic and environmental factors, which can cause significant variation, even within a single species. Multiple studies have shown that variables such as age, sex, sexual maturity, diet, temperature [31,62], and geographical origin [63] influence the CHC profiles, contributing to intra-species variation. These findings highlight the importance of accounting for these factors before CHCs can be reliably applied in forensic casework for species identification and age estimation [20].

Sexually dimorphic CHC profiles have been observed in both blow flies [35,59,64,65,66] and flesh flies [39]. Newly emerged flies of both sexes may initially appear similar in profile, however differences become more pronounced with age [67]. Trabalon et al. [64] documented changes in the CHC profile between male and female *C. vomitoria* from emergence to maturity, noting the significant difference between sexes is the presence of alkenes in females. Butterworth et al. [65] observed both quantitative and qualitative changes in CHCs of adult *Chrysomya varipes* (Macquart) over an 11-day period, with compositional changes occurring more gradually in females. Further analysis by Butterworth et al. [66], of nine Chrysomya species and one Lucilia species of Australia, found that 80% exhibited sexually dimorphic CHC profiles. Braga et al. [59] demonstrated that CHC chain lengths in *Ch. putoria* CHCs in the range C_21_–C_35_ in females and C_21_–C_37_ in males, and that females exhibited fewer age-specific compounds than males. Similarly, Barbosa et al. [35] found the CHC chain lengths in *C. macellaria* in the range C_23_–C_31_ in both sexes, while those in *H. segmentaria* were in the range C_23_–C_37_. In *L. cuprina*, females presented CHCs between C_24_ and C_35_, while males exhibited profiles ranging from C_25_ to C_35_.

Other studies have demonstrated geographical variations in CHC profiles of blow fly species relevant to forensic science, offering potential for identifying post-mortem body transport between regions. Moore et al. [63] compared the chemical profiles of *L. sericata* empty puparia from the United Kingdom (UK) and two locations in Germany (Frankfurt and Steinau, approximately 70 km apart). Although the profiles across these locations were broadly similar, specific compounds were unique to each region; for example, 9 + 11 + 13-MeC_27_ was specific to Steinau, while C_29:1_ was unique to the UK. Additionally, the *n*-alkanes C_31_, C_32_, and C_33_ were all present in UK samples; however, while C_31_ was detected in Steinau, both C_32_ and C_33_ were absent in Steinau and Frankfurt populations. These findings suggest that C_32_ and C_33_ are geographically specific rather than species-specific. Additionally, Moore et al. [63] distinguished between *C. vicina* collected from the UK, Germany, Norway, and Spain.

Further research by Kula et al. [68] compared the CHC profiles of adult and empty puparia of *C. vicina* between the UK, Germany, and Turkey, revealing that while each population could be distinguished, the UK and Germany samples were more similar compared to those from Turkey, likely reflecting their closer geographical proximity and environmental conditions. In addition to investigating chemotaxonomy across all life stages of *Ch. megacephala*, Paula et al. [37] also noted significant compositional differences between the CHCs of specimens from two locations in Brazil (Dourados and Rio Claro, approximately 870 km apart). On another scale, Kula and Moore [69] distinguished adult *L. sericata* reared indoors and outdoors using CHC analysis.

In certain forensic contexts, such as indoor environments where insect access to a cadaver is restricted and decomposition is prolonged, multiple generations of the same species may be present on decomposing remains, potentially complicating PMI estimations [70]. For instance, successive generations of *L. sericata* and *P. terraenovae* have been documented on human corpses in such settings [70]. By analysing the CHCs extracted from three generations of *Ch. megacephala*, Paula et al. [71] identified both qualitative and quantitative differences between generations (F1, F2 and F3). For example, compounds such as 5-MeC_21_ and C_33_ were only detected in F1, while C_22_ and 2-MeC_25_ were detected in F2 and F3. Additional compounds, including C_23:1_, C_27:1_, and 4,12-DiMeC_28_ were unique to F2 empty puparia.

Cuticular hydrocarbon degradation provides further insight into environmental impacts on forensic samples. Sharif et al. [52,53] examined the degradation of C_25_, C_26_, C_27_, C_28_, and C_29_ extracted from the cuticles of *L. sericata* and *C. vicina* empty puparia under differing storage conditions (paper towel vs. soil). Results indicated that soil storage significantly increased the degradation of CHC components, especially in *C. vicina* where all five CHCs fell below detectable limit after three months. Following this study, Sharif et al. [54] focused on the same five CHCs to investigate the influence of microenvironmental factors on the CHC profile of *L. sericata* empty puparia under various conditions, including indoors, outdoors above ground, and outdoors buried in soil. Conducted over three months, the study found that CHCs stored indoors remained remarkably stable, whereas significant variation was observed in the CHC components exposed to environmental factors.

These findings highlight the critical importance of considering storage conditions and the chain of custody when using CHCs in forensic entomology investigations. Variations in CHC profiles due to environmental factors or improper handling could lead to inaccurate interpretations, underscoring the need for meticulous preservation and documentation of samples throughout the forensic process.

Recently, a study found that *L. sericata* CHC profiles exhibited variability when immature flies were reared with food spiked with glyphosate [72]. This marks a novel interface between forensic entomology, toxicology (entomotoxicology) [73], and CHC analysis. These initial results offer promising insights into the potential of using cuticular chemical analysis to identify exposure to toxicological substances, providing an alternative to conventional toxicological analyses. However, this emerging area of research emphasises the need for further studies to understand how different substances, their concentrations, and species-specific variations influence CHC profiles, particularly across various developmental stages. These insights could pave the way for more comprehensive and nuanced toxicological investigations in forensic entomology.

## 3. Conclusions

This review highlights the significant advancements in the use of CHC analysis in forensic entomology since its inception in 2007, with a particular focus on research published since the most recent review in 2021. It underscores the potential of CHC profiling for species identification and age estimation, particularly in challenging cases involving empty puparia, which can be difficult to interpret using traditional methods. Despite the observed intra-species variability due to factors such as age, sex, and environmental influences, these variations can be leveraged to improve forensic investigations, such as refining PMI estimations, detecting exposure to toxicological substances, and the movement of a body across different locations. Future research should prioritise expanding geographical coverage by considering multiple locations to better understand the regional and environmental factors influencing CHC profiles. Additionally, future work should focus on exploring other factors that affect CHC composition, including environmental conditions, and the impact of toxins or drugs within the insect food source. This would further enhance the applicability of CHC analysis in forensic casework and refine its potential for global forensic entomology applications.

## Figures and Tables

**Figure 1 insects-16-00144-f001:**
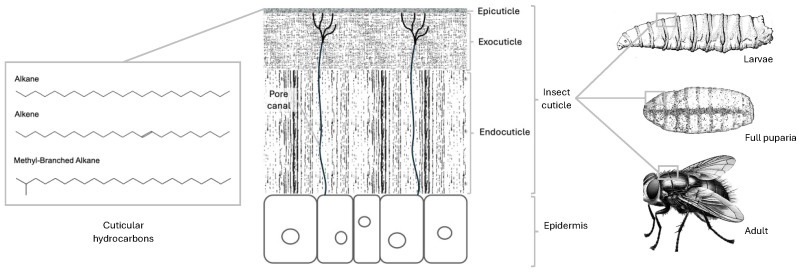
Visual representation of the insect integument and cuticular hydrocarbons. The cuticle is present at every stage of insect development (shown on the right: blow fly larvae, fully formed puparia, and adult). It is composed of three distinct layers: the epicuticle, exocuticle, and endocuticle. These layers are arranged from the outermost to the innermost part of the body, with the endocuticle in direct contact with the insect epidermis. Hydrocarbons found in the insect cuticle, which are often extracted for experimental purposes, are primarily located in the epicuticle.

## Data Availability

The data supporting this review are derived from publicly available scientific literature, as cited throughout the manuscript. Copies of the specific articles or additional details regarding the sources can be made available upon reasonable request to the corresponding authors.

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
