# Peer review of "Advances and Current Status in the Use of Cuticular Hydrocarbons for Forensic Entomology Applications"

_insects, 2025, doi:10.3390/insects16020144_

Round 1

Reviewer 1 Report

Comments and Suggestions for Authors

Dear Authors,

I would like to thank you for your work titled "Advances and current status in the use of cuticular hydrocarbons for forensic entomology applications," for which I had the opportunity to referee. Your article was truly a remarkable work with its scientific perspective, original contributions to the subject, and meticulously prepared content.

First of all, I would like to write a little about my thoughts about the manuscript.

The manuscript titled "Advances and Current Status in the Use of Cuticular Hydrocarbons for Forensic Entomology Applications" provides a comprehensive review of the applications of cuticular hydrocarbons (CHCs) in forensic entomology. This review manuscript emphasises CHC profiling's potential for species identification and age estimation in challenging forensic cases.

The outline of the manuscript is well structured, with a logical progression of ideas. Each section builds upon the previous one, offering a cohesive narrative.

The writing of the manuscript is clear.

This manuscript makes a significant contribution by summarising the latest developments in CHC research while proposing directions for future investigations.

The manuscript efficiently highlights the advancements in CHC applications and their challenges.

The extraction and analytical procedures for CHCs are described thoroughly. Figures and tables are appropriate and improve the manuscript's clarity.

This manuscript makes a significant contribution by summarising the latest developments in CHC research while proposing directions for future investigations. Its discussion on CHC weathering and toxicological interactions is noteworthy.

The authors effectively discuss environmental factors influencing CHC variability and the technique's limitations, suggesting future research directions. The manuscript is well organised. It is informative and understandable. The presented manuscript provides valuable insights into an evolving forensic methodology.

I am happy to have had the opportunity to read and evaluate such valuable work that contributes to our field.

Author Response

The authors would like to thank R1 for their thoughtful and encouraging review of our manuscript. We sincerely appreciate their kind words and detailed feedback highlighting the strengths of our work. It is truly gratifying to know that they have found the article valuable and well-structured, and we are grateful for their acknowledgment of its contribution to the field.

Their positive comments motivate us to continue advancing research in this area, and we are delighted that the manuscript resonated well with them.

Reviewer 2 Report

Comments and Suggestions for Authors

15/16

"It has been investigated the role of environmental factors (temperature, humidity, etc) on the degradation of cuticular hydrocarbons according."

improve the english; this is not proper english

16/17

"The applications of cuticular hydrocarbons in foren- sic entomology still remain"

you probably mean:

   The application of cuticular hydrocarbons in forensic entomology still remains...  

41

"whether alive or at various"

in the formatted version, there seems to be a space too much between "whether" and "alive"

50

"researched ecology and behaviour are essential for their forensic application [4]"

please add reference to historical paper → https://www.researchgate.net/publication/11883594_A_brief_history_of_forensic_entomology

69

"Within physical methods,"

do you mean "amongst"? (i am not sure)

78

the image quality is not sufficient, it looks as if it was done with a b/w photocopy machine

please redo this

92/93

"– critical considerations for investigators employing this technique."

maybe better write "we provide" / "to provide"

97

"The ep- icuticle is the outermost"

separation of the word is wrong

109

"olefins can exist as two possible isomeric forms referred to as the cis- (Z-alkenes) or trans-forms (E-alkenes)"

maybe better: exist ... in two (?)

242

"particularly for life stages that are challenging to examine by morphology such as pupae and empty puparia"

pls explain why these are particularly challenging

277

"Additionally, Moore et al. [61] provided distinction between C. vicina collected from the UK, Germany, Norway and Spain."

grammar (provided distinction) correct?

288

"In certain forensic contexts, multiple generations"

in which exactly / for example?

323

"This review highlights the significant advancements in the use of CHC analysis in forensic entomology."

since when? compared to when?

327

"these variations can be leveraged to improve forensic investigations,"

how can they be leveraged? please give examples.

Author Response

We sincerely thank Reviewer 2 for their thoughtful and constructive comments, which have helped improve the quality of our manuscript. Overall, we have accepted and addressed almost all of the comments provided. However, we would like to clarify that the concerns regarding incorrectly split words are due to a production issue that occurred when the manuscript was uploaded into the journal template. While we have corrected all instances noted by the reviewer, we would like to highlight that this issue will also be addressed during the editing process by the journal's production team.

Comment 1 (Line 15/16):

"It has been investigated the role of environmental factors (temperature, humidity, etc) on the degradation of cuticular hydrocarbons according."

improve the english; this is not proper english

Response 1:

Thank you for this comment. This section has been improved by reworking the whole section to now be “Cuticular hydrocarbons present a valuable tool in forensic entomology, aiding species identification and age estimation of necrophagous insects, particularly blow flies. This review provides a detailed overview of recent advancements in the applications of cuticular hydrocarbons, such as in post-mortem interval estimation. Additionally, it explores the factors contributing to intra-species variation, including age, sex, temperature, and geographical origin, and details how these variations can provide additional insight during legal investigations. While promising, challenges remain in the use of cuticular hydrocarbons in forensic investigations, and further research is required to enhance reliability of this method.”

Comment 2 (Line 16/17):

"The applications of cuticular hydrocarbons in foren- sic entomology still remain"

you probably mean:

The application of cuticular hydrocarbons in forensic entomology still remains...

Response 2:

Thank you for this comment. This section has been improved by reworking the whole section to now be “Cuticular hydrocarbons present a valuable tool in forensic entomology, aiding species identification and age estimation of necrophagous insects, particularly blow flies. This review provides a detailed overview of recent advancements in the applications of cuticular hydrocarbons, such as in post-mortem interval estimation. Additionally, it explores the factors contributing to intra-species variation, including age, sex, temperature, and geographical origin, and details how these variations can provide additional insight during legal investigations. While promising, challenges remain in the use of cuticular hydrocarbons in forensic investigations, and further research is required to enhance reliability of this method.”

Comment 3 (Line 41):

"whether alive or at various"

in the formatted version, there seems to be a space too much between "whether" and "alive"

Response 3:

Thank you for this comment. The double space between “whether” and “alive” has been removed.

Comment 4 (Line 50):

"researched ecology and behaviour are essential for their forensic application [4]"

please add reference to historical paper → https://www.researchgate.net/publication/11883594_A_brief_history_of_forensic_entomology

Response 4:

Thank you for this comment. The authors agree this paper holds value within this sentence and has been added as a reference.

Comment 5 (Line 69):

"Within physical methods,"

do you mean "amongst"? (i am not sure)

Response 4:

Thank you for your comment. The authors believe the word “within” is more suitable in this sentence as it is discussing techniques within the broader category of physical methods.

Comment 6 (Line 78):

the image quality is not sufficient, it looks as if it was done with a b/w photocopy machine

please redo this

Response 6:

Thank you for your comment. The quality of the original image that we produced is actually high; however, it appears to lose resolution during the process of uploading it to the journal template. To address your concern, we will provide the image as a separate high-resolution file to the editor during the resubmission process. This will ensure that the image is available in its original quality for use during the production of the paper.

Comment 7 (Line 92/93):

"– critical considerations for investigators employing this technique."

maybe better write "we provide" / "to provide"

Response 7:

Thank you for your comment. This sentence has been improved by adding “to provide” to the sentence.

Comment 8 (Line 97)

"The ep- icuticle is the outermost"

separation of the word is wrong

Response 8:

A “the” has been added to the previous sentences “The integument of an insect consists of a single epidermal cell layer and the cuticle, which is comprised of three layers: the endocuticle, exocuticle and epicuticle (Fig. 1).” This removes the separation of “ep- icuticle”.

Comment 9 (Line 109):

"olefins can exist as two possible isomeric forms referred to as the cis- (Z-alkenes) or trans-forms (E-alkenes)"

maybe better: exist ... in two (?)

Response 9:

Thank you for your comment. This sentence has been improved by changing “as” to “to”.

Comment 10 (Line 242):

"particularly for life stages that are challenging to examine by morphology such as pupae and empty puparia"

pls explain why these are particularly challenging

Response 10:

Thank you for your comment. This sentence has been improved by adding two sentences and two references.

“Inside the puparia, pupae undergo significant transformations that can only be appreciated after the puparia is carefully removed by a trained entomologist or through the application of specific technologies (REF: https://pubmed.ncbi.nlm.nih.gov/27766412/). Additionally, empty puparia often lack obvious external features, and their condition can be highly affected by weathering (REF: https://pubmed.ncbi.nlm.nih.gov/39018983/), making accurate age estimation even more difficult using traditional methods.”

Comment 11 (Line 277):

"Additionally, Moore et al. [61] provided distinction between C. vicina collected from the UK, Germany, Norway and Spain."

grammar (provided distinction) correct?

Response 11:

Thank you for your comment. To improve clarity, this sentence has been changed from "Additionally, Moore et al. [61] provided distinction between C. vicina collected from the

UK, Germany, Norway and Spain." To "Additionally, Moore et al. [61] distinguished between C. vicina collected from the UK, Germany, Norway and Spain."

Comment 12 (Line 288):

"In certain forensic contexts, multiple generations"

in which exactly / for example?

Response 12:

Thank you for this comment. In order to improve this statement, we have further expanded on this sentence providing more information and a reference “In certain forensic contexts, such as indoor environments where insect access to a cadaver is restricted and decomposition is prolonged, multiple generations of the same species may be present on decomposing remains, potentially complicating PMI estimations [69]. For instance, successive generations of L. sericata and P. terraenovae have been documented on human corpses in such settings”

Comment 13 (Line 323):

"This review highlights the significant advancements in the use of CHC analysis in forensic entomology."

since when? compared to when?

Response 13:

Thank you for your comment. This point has been expanded in both the introduction and the conclusion to clarify the rationale behind the review, as well as to highlight the advancements in CHC analysis in forensic entomology compared to previous research.

Comment 14 (Line 327):

"these variations can be leveraged to improve forensic investigations,"

how can they be leveraged? please give examples.

Response 14:

Thank you for your comment. “detecting exposure to toxicological substances” has been added to provide another example. The sentence is now: “Despite the observed intra-species variability due to factors such as age, sex, and environmental influences, these variations can be leveraged to improve forensic investigations, such as refining PMI estimations, detecting exposure to toxicological substances and the movement of a body across different locations.”

Round 2

Reviewer 2 Report

Comments and Suggestions for Authors

Please redo the figure. I already wrote that this is not a type of figure that is state of the art. Please either redraw it or use good photographs. This figure looks as if it was from a paper from the 1980s. The journal looks really bad if this would be used. I set the paper to "major" revision to stress this. You can do this, it is not a lot of work.

Author Response

Comments 1: Please redo the figure. I already wrote that this is not a type of figure that is state of the art. Please either redraw it or use good photographs. This figure looks as if it was from a paper from the 1980s.  Response 1: The Authors would like to thank Reviewer 2 for ensuring that our research is presented in the best possible way, both in terms of content and supporting figures. In the first round of review, Reviewer 2 pointed out that, and I quote, "the image quality is not sufficient, it looks as if it was done with a b/w photocopy machine, please redo this." The authors understood this comment as a concern regarding the resolution of the image, which we promptly addressed. However, in the second round of review, the focus shifted to the "style" of the image, with Reviewer 2 describing it as reminiscent of papers from the 1980s. While we respect Reviewer 2's perspective, we believe that the style originally used is appropriate for the descriptive purpose the image serves within the manuscript. It is also worth noting that the other reviewer did not raise any objections to the image. After consulting with the journal's editorial team, we were informed that there are no specific requirements for the "style" of the image, and that Reviewer 2’s comments were intended to improve the paper’s presentation rather than address the scientific content. In response, we have taken the following approach. As the figure’s purpose is to provide a clear and straightforward explanation of the concept being discussed, and as the image effectively supports the argument and aligns with the journal’s guidelines—emphasizing relevance and clarity over stylistic elements—we have retained the core of the image and made improvements to the parts that could benefit from enhancement. Specifically, we left the histological and chemical components unchanged, as they are inherently more technical in nature, but we improved the appearance of the insects. We believe this is a reasonable compromise, as further revisions based solely on the stylistic preferences of a single reviewer, after addressing the resolution concerns, would not add value to the scientific content of the manuscript. Alternatively, since the image primarily serves to visually represent information already detailed in the manuscript, we are open to removing it if that would resolve the matter.   Comments 2: The journal looks really bad if this would be used. I set the paper to "major" revision to stress this. You can do this, it is not a lot of work. Response 2: The Authors would like to raise a concern regarding Reviewer 2's understanding of the journal's guidelines. We feel that setting the paper for "major" revision based on personal stylistic preferences, without clearly identifying the original issue, is not in line with the journal’s review process.